# Difference Analysis of the Composition of Iron (Hydr)Oxides and Dissolved Organic Matter in Pit Mud of Different Pit Ages in Luzhou Laojiao and Its Implications for the Ripening Process of Pit Mud

**DOI:** 10.3390/foods12213962

**Published:** 2023-10-30

**Authors:** Kairui Jiao, Bo Deng, Ping Song, Hailong Ding, Hailong Liu, Bin Lian

**Affiliations:** 1College of Life Sciences, Nanjing Normal University, Nanjing 210023, China; jkr08061999@163.com (K.J.); liuhailong@njnu.edu.cn (H.L.); 2National Engineering Research Center of Solid State Brewing, Luzhou 646000, China; dengbo@lzlj.com (B.D.); dinghl@lzlj.com (H.D.); 3School of Food Science and Pharmaceutical Engineering, Nanjing Normal University, Nanjing 210023, China; songping@njnu.edu.cn; 4College of Marine Science and Engineering, Nanjing Normal University, Nanjing 210023, China

**Keywords:** Luzhou-flavor liquor, pit mud, DOM, iron (hydr)oxides, FT-ICR-MS

## Abstract

Long-term production practice proves that good liquor comes out of the old cellar, and the aged pit mud is very important to the quality of Luzhou-flavor liquor. X-ray diffraction, Fourier transform ion cyclotron resonance mass spectrometry, and infrared spectroscopy were used to investigate the composition characteristics of iron-bearing minerals and dissolved organic matter (DOM) in 2-year, 40-year, and 100-year pit mud and yellow soil (raw materials for making pit mud) of Luzhou Laojiao distillery. The results showed that the contents of total iron and crystalline iron minerals decreased significantly, while the ratio of Fe(II)/Fe(III) and the content of amorphous iron (hydr)oxides increased significantly with increasing cellar age. DOM richness, unsaturation, and aromaticity, as well as lignin/phenolics, polyphenols, and polycyclic aromatics ratios, were enhanced in pit mud. The results of the principal component analysis indicate that changes in the morphology and content of iron-bearing minerals in pit mud were significantly correlated with the changes in DOM molecular components, which is mainly attributed to the different affinities of amorphous iron (hydr)oxides and crystalline iron minerals for the DOM components. The study is important for understanding the evolution pattern of iron-bearing minerals and DOM and their interactions during the aging of pit mud and provides a new way to further understand the influence of aged pit mud on Luzhou-flavor liquor production.

## 1. Introduction

Chinese liquor is one of the oldest distilled spirits in the world, of which strong-flavor baijiu (SFB), also known as Luzhou-flavor liquor, is the best-selling in China, accounting for over 70% of total consumption in the Chinese baijiu market [1,2]. SFB production entails fermentation of the treated raw grains and Daqu(a kind of saccharifying and fermenting agent for Chinese liquor) in a cellar (approximate dimensions: length, 5.0 m; width, 3.0 m; and depth, 2.0 m) for 2 to 4 months [3]. The four walls and bottom of the cellar are covered with a layer of pit mud made of yellow soil [4]. Long-term production practice proves that the longer the pit mud is used, the better its quality and the higher the quality of the SFB produced by the cellar. This finding indicates that the aging of pit mud plays an important role in SFB fermentation [5]. As natural pit mud takes at least 20 years to mature, it is necessary to find ways to accelerate the maturation of new cellar and thus improve the quality of SFB. Therefore, the research into the maturation mechanism of pit mud has been a hot topic among researchers working in the domestic brewing industry.

The aging mechanism of pit mud is complicated. Previous studies focus on microbial diversity and functional analysis in pit mud. For example, Liu et al. explored the bacterial and fungal communities in pit mud of different ages and revealed the changes in microbial community structure and phylogenetic novelty therein [6,7]. Zhang et al. found significant compositional differences in bacterial community structures at different vertical depths of bottom pit mud [8]. Chai et al. studied clostridia with butyrate production potential in pit mud and elucidated the species-specific butyrate synthesis characteristics of *clostridium* [9]. These studies suggest that the microbial diversity and community structure of pit mud are closely related to the aging of pit mud. However, the study of dissolved organic matter (DOM) and mineral composition in pit mud has not been reported.

Organic matter (OM) is an important component of pit mud. Previous studies mainly focused on the determination of changes in reducing sugar, organic acid, and ammonia nitrogen in pit mud [3,10], which failed to expound all of the changes in organic components during the aging of pit mud. In addition, in terms of research methods, previous studies mainly used gas chromatography, high-performance liquid chromatography, and headspace-solid phase microextraction/gas chromatography-mass spectrometry to analyze aroma components in pit mud [3,8] and did not analyze complex organic matter components from the molecular level for pit mud. DOM is widespread in terrestrial ecosystems and is a major source of carbon and energy in soil microbial communities [11]. The pit mud contains a large amount of DOM. As the most dynamic and microbially accessible part of OM pool in pit mud, research of the variation of DOM components in pit mud will deepen the understanding of the maturation mechanism of pit mud. Fourier transform ion cyclotron resonance mass spectrometry (FT-ICR-MS) is widely used to explore the composition characteristics of complex DOM samples at the molecular level because of its ultrahigh mass accuracy and mass-resolving power [12]. Therefore, FT-ICR-MS was used in this study to analyze the composition characteristics of DOM in pit mud.

As a special form of fermented soil, the composition and structure of the minerals in pit mud have a profound influence on its physicochemical properties and microbial life activities. In the process of production practice, the periodic opening and closing of the cellar often cause a change in the redox environment in pit mud [8]. With the increase in cellar age, the color of pit mud changes from reddish to grey [10]. The active redox changes of iron were found to cause changes in the color of the soil [13]. Iron (hydr)oxides are the most important chromogenic component in soil [14]. The changes in the color of pit mud with different ages indicate that there are differences in the contents of iron and iron (hydr)oxides in pit mud, but there are no relevant reports.

In conclusion, the changes in the composition and content of DOM and iron minerals in pit mud may be an important factor affecting the aging of pit mud; therefore, in the present study, pit mud was sampled from Luzhou Laojiao distillery, a famous SFB manufacturer in China, as the research object. The chemical diversity characteristics of DOM and the composition of iron (hydr)oxides in different samples were analyzed, and the influence of their interaction on the aging of pit mud was revealed. This study helps to enrich the understanding of the maturation process of pit mud.

## 2. Materials and Methods

### 2.1. Sample Collection and Environmental Description

Samples were obtained from Luzhou Laojiao distillery, Luzhou City, Sichuan Province, China (105°29′26″ E, 28°53′51″ N, Figure 1). Raw material for making pit mud- yellow soil (unfermented, marked CK) and pit mud aged about 2, 40, and 100 years (marked 2-PM, 40-PM, and 100-PM) were selected as objects of the research. Pit mud sampling points were located at the diagonal intersection of the four walls and the bottom of the cellar. At each point, samples with a mass of 200 g were collected to a total sample mass of 1 kg. Pit mud was mixed and used as a representative sample of a cellar. Three cellars with similar ages were collected as replicates. Therefore, a total of 12 samples (4 groups × 3 replicates per group) were collected. Samples were collected using sterile sampling bags and transported to the laboratory on ice boxes. The sample was divided into two parts. One part of the fresh sample was stored in a 4 °C refrigerator for the determination of the moisture content (MC) and Fe(II), and the other part of the sample was air-dried and passed through a 100-mesh screen and then used for XRD, ATR-FTIR, FT-ICR-MS, and measurement of other physicochemical indicators.

### 2.2. Determination and Characterization of Physicochemical Properties of Samples

The MC of fresh samples was determined by drying them at 105 °C for 24 h. The pH of the samples was determined using a pH meter (PHSJ-4F, Shanghai Lei Ci, Shanghai, China) according to the soil-water ratio of 1:2.5 (*w*/*v*). The OM content of the samples was determined by the sulfuric acid-potassium dichromate oxidation method according to the Chinese National Standard NY/T 1121.6-2006 [15]. According to Chinese National Standard NY/T 1121.7-2014 [16], NH_4_F-HCl (0.03 mol/L) was used to extract available phosphorus (AP) according to the soil-water ratio of 1:10 (*w*/*v*), and then AP was determined using ion chromatograph (ICS-900, Dionex, Sunnyvale, CA, USA). Dissolved organic carbon (DOC) was determined using a total organic carbon analyzer (TOC-VCPH, Shimadzu Corporation, Kyoto, Japan). Carbonate content was determined using acid-base titration [17].

Determination of iron-related indices in samples: Fe(II) contents in fresh samples were analyzed using the ferrozine method [18]; total iron (TFe) contents were digested by mixed acid (HNO_3_-HF-HClO_4_) [19]; Fe(III) contents were calculated by subtracting Fe(II) from TFe [20]; free iron oxides (Fe_d_) were extracted using a dithionite-citrate-bicarbonate solution (0.1 mol/L) [21]; Amorphous iron oxides (Fe_o_) were extracted from ammonium oxalate solution (0.2 mol/L, pH = 3) [22]; crystalline iron oxides (Fe_c_) were calculated by subtracting Fe_o_ from Fe_d_ [23]; iron ion concentrations were measured using a flame atomic absorption spectrometer (AA-6300C, Shimadzu Corporation, Kyoto, Japan).

The mineral composition of the samples was detected using an X-ray diffractometer (XRD; XPert Pro MPD, Malvern Panalytical Corporation, Almelo, The Netherlands). The scanning angle ranged from 10 to 65°. The scanning speed was 10°/min. Organic functional groups on the sample surface were detected by attenuated total reflection Fourier transform infrared spectroscopy (ATR-FTIR; Thermo Nicolet iS5, Waltham, MA, USA). Spectral data was collected between 4000 to 400 cm^−1^. The resolution was 4000 cm^−1^. The number of scans was 32.

### 2.3. Extraction and Determination of DOM in Samples

#### 2.3.1. DOM Extraction

The air-dried samples (3.00 g) were mixed with 30 mL of Milli-Q Water (GREEN-Q2-10T, EPED, Nanjing, China) in a 50 mL centrifuge tube and then agitated in an oscillator (25 °C, 180 rpm) for 12 h. The DOM solution was obtained by centrifuging the suspension at 8000 rpm for 8 min and filtering the supernatant through a 0.45 μm filter membrane. Acidified samples were obtained by adjusting the pH of the DOM solution to 2 with concentrated hydrochloric acid (37%, ultrapure) [12].

DOM samples were enriched and desalted using a Bond Elut PPL column (500 mg, 6 mL, Agilent, Santa Clara, CA, USA). The specific procedure was described as follows [24]: 18 mL methanol (HPLC grade) and 18 mL acidified water (pH = 2.0) were used to activate the filter element. Acidified samples were passed through the filter element at a rate of about 2 mL/min under gravity. The filter element was rinsed with 18 mL of acidified water, and then the moisture in the column was dried with N_2_ (99.999% purity). Finally, 6 mL methanol (HPLC grade) was added to elute the sample. The eluent was diluted with 1:1 (*v*/*v*) methanol: water solution and then used for FT-ICR-MS analysis.

#### 2.3.2. FT-ICR-MS Analysis

The molecular composition of DOM was resolved using a Bruker solariX 94 FT-ICR-MS (Billerica, MA, USA) equipped with a 9.4-T superconducting magnet and a negative electron spray ionization source. All formulas were assigned according to the criterion of elemental combinations of C_1–100_, H_1–200_, O_0–50_, N_0–4_, S_0–2_, P. Only peaks with a signal-to-noise ratio (S/N) ≥ 6 and error threshold of ±0.5 ppm were considered for further analysis; H/C < 2.4 and O/C < 1.2 were set as further restrictions on the formula and subsequent computations. The SRNOM standard was used for calibration. The operating conditions were as follows: a 4-kV spray shield voltage, a 4.5-kV capillary column voltage, and a 320-V capillary terminal voltage. To reveal the detailed characteristics of DOM molecules, the following intensity-weighted average parameters were calculated [25]: “AI” (aromaticity index), “DBE” (double bond equivalent), “NOSC” (nominal oxidation state of carbon), “CRAMs” (carboxyl-rich alicyclic molecules), “O/C” (oxygen-to-carbon element ratios), and “H/C” (hydrogen-to-carbon element ratios).

DOM compounds were classified into six types according to AI value and ratios of H/C and O/C [26,27,28]: (1) Combustion-derived polycyclic aromatics (CDPA): AI > 0.66; (2) Vascular plant-derived polyphenols (VPDP): 0.66 ≥ AI > 0.50; (3) Highly unsaturated lignin/phenolics (HULP): AI ≤ 0.50 and H/C < 1.5; (4) Unsaturated aliphatic compounds (UAC): 2.2 ≥ H/C ≥ 1.5; (5) Saturated fatty acid (SFA): H/C ≥ 2.0 and O/C < 0.9; (6) Carbohydrates (CH): O/C: 0.67–1.2 and H/C: 1.5–2.2.

### 2.4. Statistical Analysis

The SPSS software package (Version 16.0, IBM, Armonk, NY, USA) was used for one-way analysis of variance (ANOVA) (Tukey test). Principal component analysis (PCA) was performed using Origin 2022b software (Origin Lab., Northampton, MA, USA). All results were expressed as ±SD; *p* < 0.05 was considered statistically significant.

## 3. Results and Discussion

### 3.1. Mineral Composition and Physicochemical Properties of Different Samples

The composition of iron-bearing minerals in yellow soil and pit mud samples was analyzed using XRD (Figure 2). The results showed that the mineral composition of the different samples was similar, but there were some differences in the intensity and location of the peaks. Compared with the yellow soil used for making pit mud, the peak intensity of crystalline iron-bearing minerals (such as goethite, mahlmoodite, and ferrorichterite, expressed as Fe_c_) in different pit mud samples exhibited a trend of decreasing gradually with increasing cellar age.

The results were explained as follows: (1) Iron is a sensitive redox element in geochemistry, and changes in the surrounding redox environment can alter its chemical properties [29]. In the process of SFB fermentation, with the continuous consumption of oxygen inside the cellar, the pit mud environment gradually presents a reduced state, which promotes the reduction in Fe_c_ and increases the solubility of iron, so the peak intensity of Fe_c_ gradually decreases with the increase in cellar age; (2) The weakened peak intensity of iron-bearing minerals in the XRD pattern may also be due to the partial conversion of Fe_c_ to amorphous iron (hydr)oxides. In general, long-term fermentation promotes the transformation of iron-bearing crystalline minerals and the loss of iron from minerals in pit mud. To clarify the specific changes of iron-bearing minerals, the contents of iron and iron minerals and other physical and chemical indexes in yellow soil and pit mud were determined (Table 1). The results showed that the physical and chemical properties of pit mud changed significantly with the increase in cellar age.

As shown in Table 1, the pH of the unfermented yellow soil was 6.16, which was weakly acidic, while the pH of 2-year pit mud decreased to 4.04. This was due to the large amount of lactic acid secreted by the dominant lactic acid bacteria in the low-aged pit mud [8]. As the cellar age increased, the pH of pit mud increased from 4.04 in 2-year pit mud to 6.62 in 100-year pit mud, which was mainly attributed to the degradation of organic acids such as lactic acid in pit mud and the accumulation of alkaline metabolites such as NH_4_^+^ in the long-term fermentation process [3,30]. In addition, MC increased significantly from 7.57% to 43.87%; OM increased significantly from 22.62 g/kg to 223.78 g/kg. The reason for this change was, on the one hand, the infiltration of yellow water from fermented grains into pit mud, which brought a large amount of water and OM [31], and on the other hand, the relative proportion of crystalline mineral content in pit mud decreased during the fermentation process, thus increasing the proportion of OM and MC.

During SFB fermentation, periodic cellar sealing openings can cause changes in the redox environment in pit mud, which can lead to changes in the relative contents of Fe(II) and Fe(III) (Table 1). After sealing the cellar, the environment of pit mud gradually shows a reduced state. At this time, Fe(III) in Fe_c_ was continuously decreased and dissolved to produce Fe(II) in anoxic environment [32]. As a result, the content of Fe_c_ decreased significantly from 10.93 g/kg to 1.29 g/kg, the content of Fe(II) increased significantly from 0.45 g/kg to 3.79 g/kg, and the Fe(II)/Fe(III) increased significantly from 0.01 to 0.33. In addition, the dissolved Fe(II) can be oxidized to Fe_o_ because of the increase in the dissolved oxygen level in pit mud after opening the cellar [33], so the content of Fe_o_ increased significantly from 1.51 g/kg to 7.54 g/kg. Fe_d_ included Fe_c_ and Fe_o_ [23], and the content of Fe_d_ significantly decreased with increasing cellar age (Table 1). As the cellar age increased, although the content of Fe_c_ decreased significantly and the content of Fe_o_ increased significantly, the content of Fe_d_, the sum of the two, decreased significantly (from 12.44 g/kg to 8.83 g/kg), which caused a significant decrease in the content of TFe (from 31.99 g/kg to 15.24 g/kg). This result also coincided with the result of the weakening of the peak intensity of iron minerals in the XRD pattern (Figure 2).

Duan et al. found that Fe_c_ can be tightly bound to organic carbon (OC)-derived DOC and AP, rendering this part of the OC difficult to use by microorganisms [34]. As the cellar age increased, the reductive dissolution of Fe_c_ in multiple rounds of fermentation and reduction environment resulted in the release of DOC and AP bound to Fe_c_. Therefore, from yellow soil to 100-year pit mud, the content of DOC increased significantly from 66.84 mg/L to 1202.88 mg/L, and the content of AP increased significantly from 0.05 g/kg to 13.20 g/kg. The higher content of DOC and AP in aged pit mud may improve the metabolic activity of microorganisms, which may have a positive impact on the quality of SFB products.

The carbonate content of pit mud increased significantly from 1.41% to 2.71% with increasing cellar age, which may inhibit the conversion of Fe_o_ to Fe_c_ [23]. In addition, a significant increase in DOC concentration (from 66.84 mg/L to 1202.88 mg/L) could retard or inhibit the conversion of Fe_o_ to Fe_c_ [35]. In general, these results led to lower amounts of crystalline iron minerals and more amorphous iron (hydr)oxides in aged pit mud.

### 3.2. Changes in DOM Molecular Composition during Aging of Pit Mud

To compare the differences in chemical properties of DOM molecules in different samples, the intensity-weighted average parameters of DOM molecules were calculated (Table 2). The results showed that: with the increase in cellar age, (1) The DOM molecular formula number increased from 3636 to 7858, indicating that the richness of DOM molecule increased; (2) The DBE value increased from 3.20 to 6.65, indicating that unsaturated molecules rich in double bonds and/or rings gradually accumulated in DOM molecules, so the unsaturation of DOM molecules increased and the H/C ratios decreased correspondingly (from 1.94 to 1.43); (3) The AI value increased from 0.05 to 0.16, indicating the accumulation of aromatic compounds in DOM molecules; (4) The NOSC value changed from −1.44 to −0.55, implying the increase in oxidized forms of carbons in DOM molecules [36]. (5) CRAMs increased from 572 to 2616, showing the accumulation of difficult-to-degrade carboxyl-rich alicyclic compounds in DOM molecules [36]; (6) The O/C ratios increased from 0.22 to 0.28, indicating that the degree of oxidation of DOM molecules increased, which was also consistent with increasing NOSC values in Table 2.

The data in Table 2 were also supported by the results of ATR-FTIR spectral analysis. As shown in Figure 3, the ATR-FTIR spectra of all samples exhibited similar absorption peak positions, but the relative intensity of the peaks differed. The absorption peaks of the main organic functional groups of the samples were presented as follows: (1) The characteristic peaks at 1430 and 1535 cm^−1^ were assigned to the C-O symmetric stretch in the COO- of carboxylic acids [37]; (2) The characteristic peak at 1630 cm^−1^ was assigned to the C=C stretching in the aromatic group and the COO- asymmetric stretching vibration in the carboxyl group [38]; (3) The characteristic peaks at 2800 to 3500 cm^−1^ were assigned to aromatic, carboxylic and phenolic groups [39]. The enhancement of these peaks further showed that compounds rich in aromatic, carboxyl, and phenolic groups were present in increasing concentrations during the aging of pit mud.

In general, increasing cellar age significantly changed the molecular diversity and chemical properties of DOM. DOM in aged pit mud had higher DBE values, AI values, and more CRAMs (Table 2). This indicates that compounds with high unsaturation and aromaticity, as well as alicyclic compounds rich in carboxyl groups, gradually dominate as fermentation proceeds. The accumulation of these compounds may result from changes in microbial metabolites during the aging process and the accumulation of stubborn organic compounds such as highly unsaturated and aromatic compounds [40].

The classification of DOM compounds in different samples was based on H/C, O/C ratios, and AI values (Figure 4A) [26]. The results showed that there were significant differences in the types and molecular compositions of DOM compounds in yellow soil and pit mud. Saturated fatty acids (60.38%) accounted for the largest proportion in yellow soil, followed by unsaturated aliphatic compounds (31.42%). Unsaturated aliphatic compounds (56.69–68.53%) dominated in pit mud, followed by lignin (14.32–22.27%) and saturated fatty acids (15.06–18.47%). As shown in Figure 4A, the proportions of lignin, polyphenols, and polycyclic aromatics gradually increased with cellar age due to the stubborn nature of these compounds, which were not easily degraded by microorganisms [37]. In addition, these stubborn compounds may also be derived from exogenous plant residues (such as sorghum hulls and wheat hulls) during fermentation [40]. Compared with yellow soil, the proportion of unsaturated aliphatic compounds increased significantly, while the proportions of saturated fatty acids and carbohydrates decreased significantly. Unsaturated aliphatic compounds are usually derived from bacterial metabolites [39]. This suggests that, as fermentation proceeds, the bacterial diversity in pit mud has become significantly different from that of the yellow soil, and the bacteria in pit mud may be more inclined to produce more unsaturated aliphatic compounds. As carbohydrates are more bio-unstable and most easily utilized by microorganisms [40], the proportion of carbohydrates in pit mud was reduced from 0.19% to 0.06% (Figure 4A). Carbohydrates are more biologically unstable and most easily utilized by microorganisms [37], so the reduction in the proportion of carbohydrates in pit mud from 0.19% to 0.06% can be ascribed to the consumption of microbial metabolic activities (Figure 4A).

DOM compounds can also be further divided into different molecular types based on their element composition (Figure 4B) [26]. The results indicated that yellow soil had the largest proportion of CHOS (42.83%) molecules, followed by CHONP (18.07%) molecules and CHON (13.79%) molecules. Compared with yellow soil, CHOP (47.77–54.60%) molecules and CHO (17.03–20.28%) molecules increased greatly and dominated in pit mud, while CHON (5.33–9.65%), CHONP (3.98–9.0%) and CHOS (3.58–4.28%) molecules decreased greatly. Studies have shown that sulfur-containing compounds are widely applied in the production of detergents and personal care products [41]. Therefore, the high proportion of CHOS molecules in yellow soil indicated that yellow soil may be affected by human activities to some extent.

DOM is widespread in terrestrial ecosystems and plays a critical role in the evolution of soil biogeochemical processes and structure [11]. The large differences in compound type and elemental composition of DOM in yellow soil and pit mud reflect the complex biochemical transformation processes that occur in pit mud during SFB fermentation. This process may have a potentially beneficial effect on the quality of SFB. Further research will provide a theoretical basis for promoting the aging of pit mud and selecting suitable raw materials for yellow soil.

### 3.3. Association of DOM Molecules with Iron (Hydr)Oxides

PCA was used to evaluate the correlation between iron (hydr)oxides and DOM compounds (Figure 5). The different samples were clearly separated in the PC1 and PC2 axes. The PC1 and PC2 axes explained 75.3% and 10.3% of the total variance, respectively.

The carboxyl groups of aromatic DOM molecules can undergo ligand exchange with the hydroxyl groups on the surface of iron hydroxide, and thus DOM with aromatic structures (such as CDPA, VPDP, and HULP) are more susceptible to bonding with iron (hydr)oxides to form Fe-OM complexes [42,43]. In addition, Fe_o_ has a larger specific surface area and more binding sites compared to Fe_c_, resulting in a stronger adsorption capacity to preferentially bind to CDPA, VPDP, and HULP [44,45]. Thus, CDPA, VPDP, and HULP were negatively correlated with Fe_c_ and positively correlated with Fe_o_ (Figure 5, Appendix A). In addition, OM is highly chemically reactive towards iron (hydr)oxides and can affect the transformation of their morphology [46]. CDPA, VPDP, and HULP can block the reduction in Fe_o_ upon binding to Fe_o_, thereby delaying or inhibiting the conversion of Fe_o_ to Fe_c_ [19]. Thus, there were more Fe_o_ and less Fe_c_ in aged pit mud (Table 1).

Correlation analysis of NOSC values, O/C ratios, and Fe(II)/Fe(III) can also further explain the relationship between DOM and iron (hydr)oxides. The NOSC value and O/C ratios can represent the oxidation state of DOM molecules [36]. The continuous oxidation of DOM molecules (supplying energy to the fermentation process) led to an increase in the degree of oxidation of DOM molecules, and therefore, the NOSC values and O/C ratios of DOM molecules increased (Table 1). The electrons generated due to the oxidation of DOM molecules can be transferred to Fe(III) on the surface of iron (hydr)oxides to form Fe(II) [44], leading to an increase in Fe(II)/Fe(III). Thus, the NOSC values and O/C ratios were positively correlated with Fe(II)/Fe(III) (Figure 5, Appendix A).

AP was negatively correlated with Fe_c_ and CHONP molecules (Figure 5, Appendix A), indicating that with the increase in cellar age, the increase in the content of AP in pit mud may be due to the release of originally adsorbed phosphoric acid from the reduction in total Fe_c_ on the one hand, and the transformation of CHONP compounds on the other hand. In addition, CHOP compounds were negatively correlated with CHONP compounds (Figure 5, Appendix A), suggesting that the increase in CHOP compounds may be attributed to the conversion of CHONP compounds.

To sum up, as the cellar age increased, the compositional characteristics of the iron-bearing minerals and DOM in pit mud changed significantly. Crystalline iron minerals predominate in young pit mud, while amorphous iron (hydr)oxides are more dominant in aged pit mud. The richness, unsaturation, and aromaticity of DOM increased significantly during the aging of pit mud. DOM from aged pit mud contained more lignin/phenolics, polyphenols, and polycyclic aromatics. Furthermore, there is a significant correlation between the changes in the morphology and content of iron-bearing minerals in pit mud and the chemical characterization of the DOM molecules. Therefore, the use of microbial fermentation and other biotechnological methods to reduce the iron content of local yellow soil (raw material for pit mud production) may help to promote the aging of new pit mud.

## 4. Conclusions

In summary, the contents of total iron and crystalline iron minerals decreased significantly with the increase in cellar age, and the ratio of Fe(II)/Fe(III) and the content of amorphous iron (hydr)oxides increased significantly. In connection with this, the richness, unsaturation, and aromaticity of DOM and the ratio of lignin/phenolics, polyphenols, and polycyclic aromatics in pit mud were increased. The research showed that molecular diversity and chemical properties of DOM compounds were closely correlated with the content and morphological changes of iron (hydr)oxides during the aging process of pit mud, and the interaction between DOM and iron (hydr)oxides changes the inorganic and organic components in pit mud, which further changes the physicochemical properties of pit mud and affects the aging of pit mud. The changes in the content and composition of iron-bearing minerals and DOM are important potential factors affecting the aging of pit mud, which not only provides a new idea to improve the aging of pit mud by using biotechnology but also helps to understand the process and mechanism of aging of pit mud from the perspective of mineral and DOM composition.

## Figures and Tables

**Figure 1 foods-12-03962-f001:**
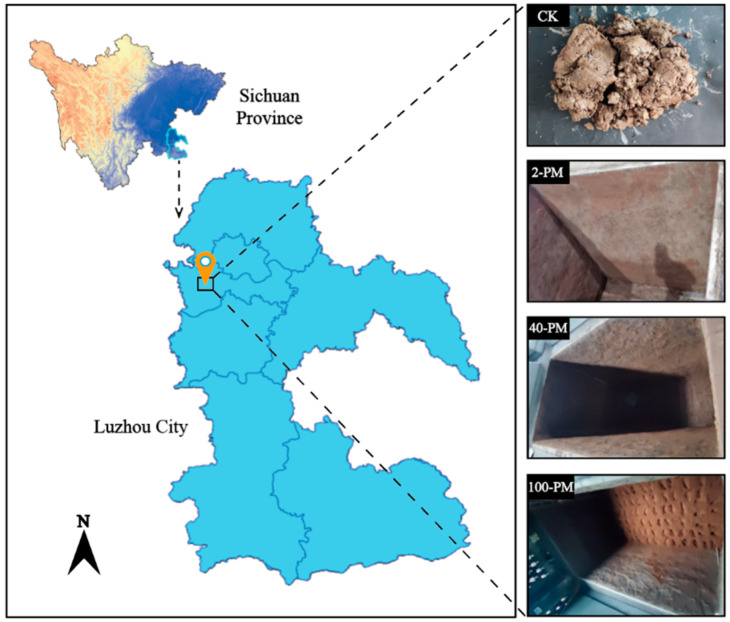
Location map of sampling sites in Luzhou, Sichuan. CK, yellow soil; 2-PM, 2-year pit mud; 40-PM, 40-year pit mud; 100-PM, 100-year pit mud.

**Figure 2 foods-12-03962-f002:**
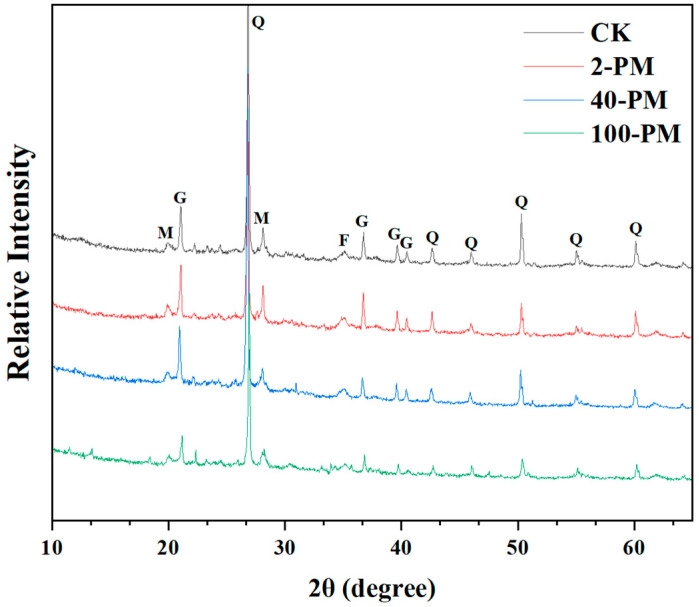
The mineral composition varies in different samples. CK, yellow soil; 2-PM, 2-year pit mud; 40-PM, 40-year pit mud; 100-PM, 100-year pit mud; Q, quartz; G, goethite; M, mahlmoodite; F, ferrorichterite.

**Figure 3 foods-12-03962-f003:**
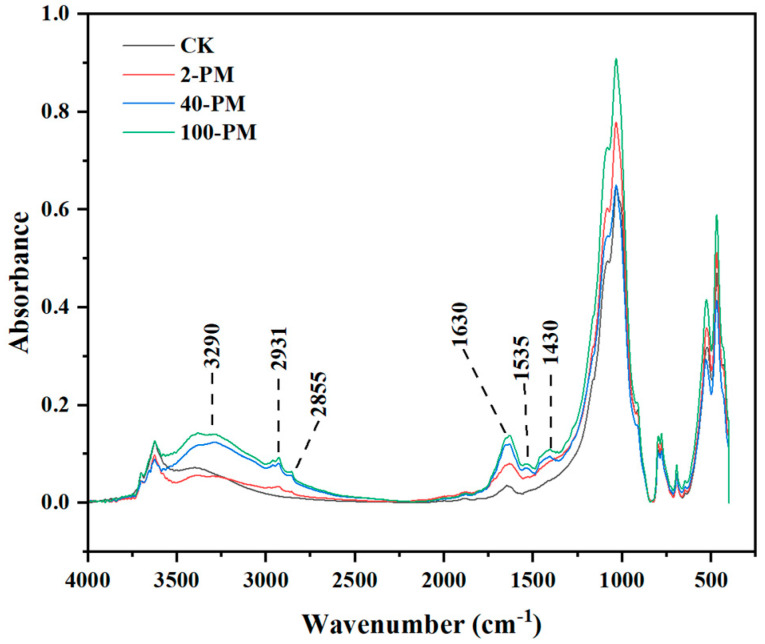
ATR-FTIR spectra of different samples. CK, yellow soil; 2-PM, 2-year pit mud; 40-PM, 40-year pit mud; 100-PM, 100-year pit mud.

**Figure 4 foods-12-03962-f004:**
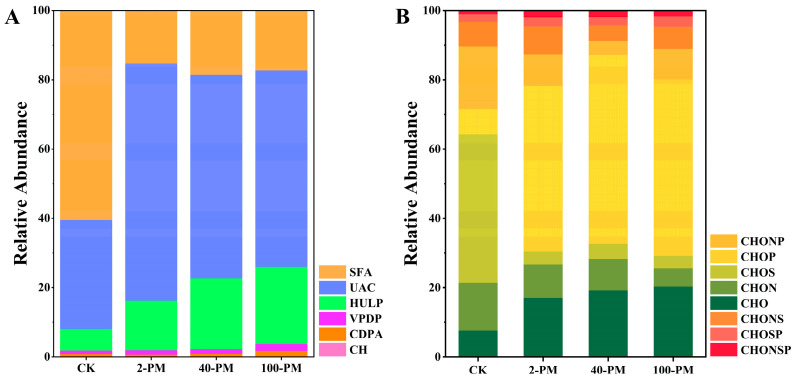
Type of DOM compounds (**A**) and element composition (**B**). SFA, saturated fatty acid; UAC, unsaturated aliphatic compounds; HULP, highly unsaturated lignin/phenolics; VPDP, vascular plant-derived polyphenols; CDPA, combustion-derived polycyclic aromatics; CH, carbohydrates. CHO, any formulae without heteroatoms; CHON, any formulae with N only; CHOS, any formulae with S only; CHOP, any formulae with P only; CHONS, any formulae with N and S; CHONP, any formulae with N and P; CHOSP, any formulae with S and P; CHONSP, any formulae with N, S, and P. CK, yellow soil; 2-PM, 2-year pit mud; 40-PM, 40-year pit mud; 100-PM, 100-year pit mud.

**Figure 5 foods-12-03962-f005:**
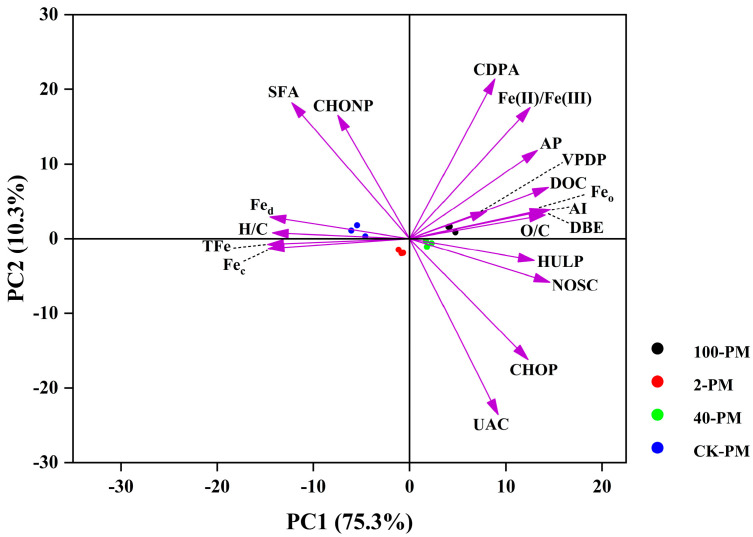
DOM molecular characteristics of different samples and PCA analysis of iron (hydr)oxides. CK, yellow soil; 2-PM, 2-year pit mud; 40-PM, 40-year pit mud; 100-PM, 100-year pit mud. AP, available phosphorus; DOC, dissolved organic carbon; TFe, total iron; Fe_d_, free iron oxides; Fe_o_, amorphous iron oxides; Fe_c_, crystalline iron oxides; Fe(II)/Fe(III), ratio of divalent to trivalent iron ions; H/C, hydrogen-to-carbon element ratios; O/C, oxygen-to-carbon element ratios; DBE, double bond equivalence; AI, aromaticity index; NOSC, the nominal oxidation state of carbon; HULP, highly unsaturated lignin/phenolics; VPDP, vascular plant-derived polyphenols; CDPA, combustion-derived polycyclic aromatics; UAC, unsaturated aliphatic compounds; SFA, saturated fatty acid; CHONP, any formulae with N and P; CHOP, any formulae with P only.

**Table 1 foods-12-03962-t001:** Physicochemical properties of yellow soil, 2-year, 40-year, and 100-year pit mud.

Parameter	CK	2-PM	40-PM	100-PM
pH	6.16 ± 0.04 ^b^	4.04 ± 0.03 ^d^	5.10 ± 0.22 ^c^	6.62 ± 0.11 ^a^
MC (%)	7.57 ± 0.33 ^d^	23.71 ± 1.53 ^c^	32.60 ± 3.28 ^b^	43.87 ± 1.68 ^a^
OM (g/kg)	22.62 ± 1.74 ^d^	89.14 ± 4.72 ^c^	168.99 ± 26.21 ^b^	223.78 ± 6.95 ^a^
AP (g/kg)	0.05 ± 0.00 ^c^	0.30 ± 0.02 ^c^	10.54 ± 0.40 ^b^	13.20 ± 0.36 ^a^
DOC (mg/L)	66.84 ± 3.38 ^d^	381.58 ± 18.07 ^c^	828.23 ± 91.26 ^b^	1202.88 ± 78.97 ^a^
Carbonate (%)	3.55 ± 0.28 ^a^	1.41 ± 0.13 ^c^	2.33 ± 0.11 ^b^	2.71 ± 0.30 ^b^
TFe (g/kg)	31.99 ± 0.50 ^a^	24.26 ± 1.55 ^b^	20.21 ± 0.18 ^c^	15.24 ± 0.28 ^d^
Fe(II) (g/kg)	0.45 ± 0.03 ^d^	0.74 ± 0.05 ^c^	2.04 ± 0.07 ^b^	3.79 ± 0.13 ^a^
Fe(III) (g/kg)	31.54 ± 0.53 ^a^	23.52 ± 1.53 ^b^	18.17 ± 0.19 ^c^	11.44 ± 0.33 ^d^
Fe(II)/Fe(III)	0.01 ± 0.00 ^c^	0.03 ± 0.01 ^c^	0.11 ± 0.01 ^b^	0.33 ± 0.02 ^a^
Fe_d_ (g/kg)	12.44 ± 0.22 ^a^	10.52 ± 0.32 ^b^	9.20 ± 0.25 ^c^	8.83 ± 0.05 ^c^
Fe_o_ (g/kg)	1.51 ± 0.22 ^d^	3.69 ± 0.04 ^c^	6.16 ± 0.07 ^b^	7.54 ± 0.51 ^a^
Fe_c_ (g/kg)	10.93 ± 0.31 ^a^	6.83 ± 0.29 ^b^	3.03 ± 0.27 ^c^	1.29 ± 0.51 ^d^

Note: MC, the moisture content; OM, organic matter; AP, available phosphorus; DOC, dissolved organic carbon; TFe, total iron; Fe_d_, free iron oxides; Fe_o_, amorphous iron oxides; Fe_c_, crystalline iron oxides; CK, yellow soil; 2-PM, 2-year pit mud; 40-PM, 40-year pit mud; 100-PM, 100-year pit mud. Values are means ± SDs of three replicates. Different small letters in the same column represent significant differences at the 0.05 level.

**Table 2 foods-12-03962-t002:** Intensity-weighted average parameters of DOM.

Sample	Formula Number	O/C	H/C	DBE	AI	NOSC	CRAMs
CK	3636 ± 227 ^c^	0.22 ± 0.01 ^c^	1.94 ± 0.08 ^a^	3.20 ± 0.13 ^c^	0.05 ± 0.03 ^c^	−1.44 ± 0.09 ^d^	572 ± 79 ^c^
2-PM	5459 ± 901 ^b^	0.25 ± 0.01 ^bc^	1.67 ± 0.04 ^b^	4.45 ± 0.14 ^b^	0.10 ± 0.01 ^b^	−0.88 ± 0.04 ^c^	1419 ± 532 ^b^
40-PM	6762 ± 294 ^ab^	0.27 ± 0.01 ^ab^	1.53 ± 0.02 ^bc^	5.38 ± 0.32 ^b^	0.13 ± 0.01 ^ab^	−0.69 ± 0.01 ^b^	2514 ± 142 ^a^
100-PM	7858 ± 224 ^a^	0.28 ± 0.16 ^a^	1.43 ± 0.11 ^c^	6.65 ± 0.87 ^a^	0.16 ± 0.01 ^a^	−0.55 ± 0.01 ^a^	2616 ± 164 ^a^

Note: Formula number, the number of DOM molecular formulas identified; DBE, double bond equivalence; AI, aromaticity index; NOSC, the nominal oxidation state of carbon; CRAMs, carboxyl-rich alicyclic molecules. CK, yellow soil; 2-PM, 2-year pit mud; 40-PM, 40-year pit mud; 100-PM, 100-year pit mud. Values are means ± SDs of three replicates. Different small letters in the same column represent significant differences at the 0.05 level.

## Data Availability

The data used to support the findings of this study can be made available by the corresponding author upon request.

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
