# Peer review of "Difference Analysis of the Composition of Iron (Hydr)Oxides and Dissolved Organic Matter in Pit Mud of Different Pit Ages in Luzhou Laojiao and Its Implications for the Ripening Process of Pit Mud"

_foods, 2023, doi:10.3390/foods12213962_

Round 1

Reviewer 1 Report

Comments and Suggestions for Authors

1) Introduction - in my opinion the information about the used methods for pit mud determination should be added.

2) Analysis correlation should be performed and showed results in the table.

3) Is the iron the most important element in pit mud? Why did not you determine other elements?

4) Conclusion should be corrected because of lack of results. 

5) How many samples did you analyzed?

Author Response

Response to Reviewer 1 Comments

Thank you very much for taking the time to review this manuscript. Now, we have accepted all the suggestions, and revised it point-by-point (The modified part is in the revised manuscript with track change). We hope our answers can satisfy with you. Thank you very much for your valuable comments, which greatly indeed improve the quality of this article.

Comments 1: Introduction - in my opinion the information about the used methods for pit mud determination should be added.

Response 1: Thanks for your suggestion. We have added more information on the method of determination of pit mud. (1) In addition, in terms of research methods, previous studies mainly used gas chromatography, high-performance liquid chromatography, and headspace-solid phase microextraction/gas chromatography-mass spectrometry to analyze aroma components in pit mud [3,8], and did not analyze complex organic matter components from the molecular level for pit mud. Please see this from line 58 to 62 in the new revised manuscript. (2) Fourier transform ion cyclotron resonance mass spectrometry (FT-ICR-MS) is widely used to explore the composition characteristics of complex DOM samples at the molecular level because of its ultrahigh mass accuracy and mass-resolving power[21]. Therefore, FT-ICR-MS was used to analyze the composition characteristics of DOM in pit mud. Please also see this from line 66 to 70 in the new revised manuscript.

Comments 2: Analysis correlation should be performed and showed results in the table.

Response 2: We agree with your suggestion. We have supplemented the correlation analysis and presented the results in Table S1, which was uploaded as an attachment. Correspondingly, we add "Table S1" in lines 352, 365, 366, and 371. Thank you again for your constructive comments.

Comments 3: Is the iron the most important element in pit mud? Why did not you determine other elements?

Response 3: Thanks for your comments. Iron was determined as an important element for the following reasons: (1) It was observed that there were differences in the color of pit mud at different ages, the color of younger pit mud was red, and the color of older pit mud was gray. Previous studies have shown that soil color is closely related to the content of iron and iron minerals, but the rule of iron change in the aging process of pit mud has not been reported. (2) In the production process of Luzhou-flavor liquor, it is accompanied by periodic opening and sealing of cellars, which will lead to changes in the valence state and content of elements sensitive to the redox environment, such as iron, in pit mud. Based on the above reasons, we focus on the determination of iron related indicators.

Comments 4: Conclusion should be corrected because of lack of results.

Response 4: Thanks for your suggestions. We have corrected our conclusion as follows: In summary, the contents of total iron and crystalline iron minerals decreased significantly with the increase of cellar age, and the ratio of Fe(Ⅱ)/Fe(Ⅲ) and the content of amorphous iron (hydr)oxides increased significantly. In connection with this, the richness, unsaturation and aromaticity of DOM and the ratio of lignin/phenols, polyphenols and polycyclic aromatics in pit mud were increased. The research showed that molecular diversity and chemical properties of DOM compounds were closely correlated with the content and morphological changes of iron (hydr)oxides during the aging process of pit mud, and the interaction between DOM and iron (hydr)oxides changes the inorganic and organic components in pit mud, which further changes the physicochemical properties of pit mud and affects the aging of pit mud. The changes in the content and composition of iron-bearing minerals and DOM are important potential factors affecting the aging of pit mud, which not only provides a new idea to improve the aging of pit mud by using biotechnology, but also helps to understand the process and mechanism of aging of pit mud from the perspective of mineral and DOM composition.

Comments 5: How many samples did you analyzed?

Response 5: Thanks for your comments. We analyzed four groups of samples (yellow soil, 2-year pit mud, 40-year pit mud and 100-year pit mud). Each group of samples had three replicates. Thus, a total of 12 samples were analyzed.

 Please see the attachment. The attachment include Table S1.

Reviewer 2 Report

Comments and Suggestions for Authors

´
Revised paper reports a study of the effect of nature of organic matter and iron hydroxides present in pit mud employed for elaboration of Luzhou Laojiao licquor, and how the age of the pit mud affects the final product. 4 age types of Pit muds were analyzed for Fe(II) content, Fe(III), organic matter, TOC, phosphate, carbonate, plus additional multispecies instrumental techniques.

The study is sound, well conducted and discussed, and presents interesting results, and valuable insights in the physico-chemical processes intervening in the mud aging. Only, there are a number of considerarions, to eliminate typos and errors, and to clarify experimental details that authors should consider:

Is Luzhous Laojiao a company producing the licquor named Baijiu? If this is the case, it should be replaced by the latter in the title. Is very awkward to have a research paper stating a commercial company in its title.

lines 14,121,  Fourier must go capitalized, as it refers to a person.

line 20 - DOM richness is extrange term - Do authors mean DOM amount?

line 35 - Give an alternate name to Daqu for a non chinese reader.

line 110 - I think ferrozine method produces a red-violet color, not UV.

line 119 - Verify manufacturer. ISn't it Malvern?

line 145 - "C1-100H1-200O0-50N0-4S0-2P", make this part more readable separating atoms by commas.

lines 196, 260 - describe meaning intended for a,b,c,d

lines 218 - 240 - Use subindex in the naming of Fed, Feo and Fec, as done previously on TAble 1.

line 236 - Any idea in which form is the carbonate? Calcium carbonate, magnesium carbonate, ferric carbonate, ...

line 245 - Richness of DOM molecule. VEry awkward terminology. Do authors mean average Molecular Weight of the matter present?

line 251 - Probably the decrease of the NOSC index indicates an increase of oxidized forms of carbons, rather than carbon oxide itself.

Fig 3 - Why do the authors use a plot of absorbance in this Figure. The standard form of representing IR spectra is with transmitance. Or perhaps there is a special need to show some special detail?

Fig 5 - The loadings part in this biplot, sketched in blue color, distorts interpretation of the clusters from the scores plot (as one of the custers is also blue). Separate better colors of the loadings and scores plots.

Comments on the Quality of English Language

English has many typos, naming out of conventions, and other minor issues. Reading and understanding is reasonable.

Author Response

Response to Reviewer 2 Comments

Thank you very much for taking the time to review this manuscript. Now, we have accepted all the suggestions, and revised it point-by-point (The modified part is in the revised manuscript with track change). We hope our answers can satisfy with you. Thank you very much for your valuable comments, which greatly indeed improve the quality of this article.

Comments 1: Is Luzhous Laojiao a company producing the licquor named Baijiu? If this is the case, it should be replaced by the latter in the title. Is very awkward to have a research paper stating a commercial company in its title.

Response 1: Thank you for your comments. “Luzhou Laojiao” is a typical representative of Luzhou-flavor liquor production enterprises in China, so when people see the term “Luzhou Laojiao”, they can be associated with China's Luzhou-flavor liquor. Therefore, adding “Luzhou Laojiao” in the title can better highlight the research theme. In addition, many published papers have included the word "Luzhou Laojiao" in their titles. For example, A 2017 paper published in volume 102 of the journal “Food Research International” was titled "Deep sequencing reveals high bacterial diversity and phylogenetics. novelty in pit mud from Luzhou Laojiao cellars for Chinese strong-flavor Baijiu ".

Comments 2: lines 14,121, Fourier must go capitalized, as it refers to a person.

Response 2: “fourier” has been corrected to “Fourier”. Thanks !

Comments 3: line 20 - DOM richness is extrange term - Do authors mean DOM amount?

Response 3: DOM richness means DOM amount and diversity. In addition, “richness” is often used to describe DOM. For example, as shown in the image below. Thanks !

Comments 4: line 35 - Give an alternate name to Daqu for a non chinese reader.

Response 4: Thanks for your suggestions. "Daqu "has been corrected to "Daqu (a kind of saccharifying and fermenting agent for Chinese liquor)". The change can be found from 35 to 36 lines in the new revised manuscript.

Comments 5: line 110 - I think ferrozine method produces a red-violet color, not UV.

Response 5: “the ferrozine ultraviolet absorbance method” has been changed to “the ferrozine method”. The change can be found on line 118 in the new revised manuscript. Thanks for your suggestions!

Comments 6: line 119 - Verify manufacturer. Isn't it Malvern?

Response 6: Thanks for your suggestions. We have confirmed that the manufacturer is indeed Malvern. The manufacturer has been corrected to Malvern Panalytical Corporation. The instrument manufacturing country is changed to HL accordingly. The change can be found on line 126 in the new revised manuscript.

Comments 7: line 145 - "C1-100H1-200O0-50N0-4S0-2P", make this part more readable separating atoms by commas.

Response 7: Thanks for your suggestions. “C1-100H1-200O0-50N0-4S0-2P” has been changed to “C1-100, H1-200, O0-50, N0-4, S0-2, P”. The change can be found on line 151 in the new revised manuscript.

Comments 8: lines 196, 260 - describe meaning intended for a,b,c,d

Response 8: We have added the following description: Different small letters in the same column represent significant differences at 0.05 level. The change can be found on line 203 and 266 in the new revised manuscript. Thanks for your suggestions.

Comments 9: lines 218 - 240 - Use subindex in the naming of Fed, Feo and Fec, as done previously on Table 1.

Response 9: Thanks for your suggestions. We have corrected “Fed, Feo and Fec” to “Fed, Feo and Fec”. The change can be found from line 219 to line 240 in the new revised manuscript. In addition, we have corrected “Fed, Feo and Fec” elsewhere in the paper accordingly.

Comments 10: line 236 - Any idea in which form is the carbonate? Calcium carbonate, magnesium carbonate, ferric carbonate, ...

Response 10: Thanks for your comments. We have compared the XRD results with the XRD database, but did not find a match for the diffraction peaks of the specific carbonate. Therefore, the carbonate we determined using acid-base titration should be amorphous carbonate.

Comments 11: line 245 - Richness of DOM molecule. Very awkward terminology. Do authors mean average molecular weight of the matter present?

Response 11: DOM richness means DOM amount and diversity. Please also reference our answer to Comments 3. Thanks for your comments!

Comments 12: line 251 - Probably the decrease of the NOSC index indicates an increase of oxidized forms of carbons, rather than carbon oxide itself.

Response 12: We agree with this suggestion, “carbon oxide” has been corrected to “oxidized forms of carbons”. The change can be found on line 257. Thanks for your suggestions.

Comments 13: Fig 3 - Why do the authors use a plot of absorbance in this Figure. The standard form of representing IR spectra is with transmitance. Or perhaps there is a special need to show some special detail?

Response 13: Thanks for your comments. We don't have a special need to show some special details. We use Attenuated Total Reflection Fourier Transform Infrared Spectroscopy (ATR-FTIR), which is based on FTIR with the addition of an ATR accessory. The absorption spectrum of the sample can be obtained by absorbing the attenuated total reflection wave. Therefore, when the data measured by ATR-FTIR method is plotted, the vertical coordinate is expressed in terms of absorbance.

Comments 14: Fig 5 - The loadings part in this biplot, sketched in blue color, distorts interpretation of the clusters from the scores plot (as one of the custers is also blue). Separate better colors of the loadings and scores plots.

Response 14: Thanks for your suggestions. The loading part of the biplot has been changed to be sketched in purple for a better presentation. The corrected biplot can be found on line 339. Thank you again for your valuable suggestions. In addition, in order to better understand this graph, we also added the following notes: HULP, highly unsaturated lignin/phenolics; VPDP, vascular plant-derived polyphenols; CDPA, combustion-derived polycyclic aromatics; UAC,unsaturated aliphatic compounds; SFA: saturated fatty acids.

Response to Comments on the Quality of English Language

Point 1: English has many typos, naming out of conventions, and other minor issues. Reading and understanding is reasonable.

Response 1: Thanks for your comments. We have made corrections for typos, naming out of conventions and minor issues. Thank you again for your valuable comments.

Reviewer 3 Report

Comments and Suggestions for Authors

Paper is well written and text is clear and easy to read, with a large amount of detailed information. Title and abstract actually reflect what is covered in the review. Abstract are succinct and comprehensible and figures and tables are understandable and readable.

There are minor grammatical errors that authors should pay attention to:

line 28. there are two points left

line 104. NH4F – subscript

line 111. (HNO3-HF-HClO4) - subscript

line 137. N2 – subscript

line 205. NH4+ – superscript

line 266, 267, 269. cm-1 - subscript

line 311. Figure 4B instead Figure 4A

Comments on the Quality of English Language

Language and phrasing is clear and unambiguous to avoid confusion.

Author Response

Response to Reviewer 3 Comments

Thank you very much for taking the time to review this manuscript. Now, we have accepted all the suggestions, and revised it point-by-point (The modified part is in the revised manuscript with track change). We hope our answers can satisfy with you. Thank you very much for your valuable comments, which greatly indeed improve the quality of this article.

Comments 1: line 28. there are two points left

Response 1: We have delete an extra point. Thank you!

Comments 2: line 104. NH4F – subscript

Response 2: We have changed NH4F to NH4F. Thanks a lot!

Comments 3: line 111. (HNO3-HF-HClO4) - subscript

Response 3: We have changed HNO3-HF-HClO4 to HNO3-HF-HClO4. Thank you!

Comments 4: line 137. N2–subscript

Response 4: We have changed N2 to N2. Thank you!

Comments 5: line 205. NH4+ – superscript

Response 5: Thank you for pointing this out. We have changed NH4+ to NH4+. Thank you!

Comments 6: line 266, 267, 269. cm-1 - subscript

Response 6: We have changed cm-1 to cm-1. In addition, we have also corrected cm-1 on line 129 to cm-1. Thank you very much.

Comments 7: line 311. Figure 4B instead Figure 4A

Response 7: Thank you, we have changed it.

Response to Comments on the Quality of English Language

Point 1: Language and phrasing is clear and unambiguous to avoid confusion.

Response 1: Thanks for your comments. We have made corrections for typos, naming out of conventions and minor issues.
